# Inflammation, Anxiety, and Stress in Attention-Deficit/Hyperactivity Disorder

**DOI:** 10.3390/biomedicines9101313

**Published:** 2021-09-24

**Authors:** Luigi F. Saccaro, Zoé Schilliger, Nader Perroud, Camille Piguet

**Affiliations:** 1Clinical Neuroscience Department, Geneva University Hospital, 1205 Geneva, Switzerland; luigifrancesco.saccaro@hcuge.ch; 2Center for Psychiatric Neuroscience, Department of Psychiatry, Centre Hospitalier Universitaire Vaudois and University of Lausanne (CHUV-UNIL), 1008 Prilly-Lausanne, Switzerland; zoe.schilliger@unil.ch; 3Psychiatry Department, Faculty of Medicine, University of Geneva, 1201 Geneva, Switzerland; nader.perroud@hcuge.ch; 4Child and Adolescence Psychiatry Division, Geneva University Hospital, 1205 Geneva, Switzerland; 5Department of Psychiatry, Faculty of Medicine, University of Geneva, Campus Biotech, 9 ch des Mines, 1202 Geneva, Switzerland

**Keywords:** emotion dysregulation, affective lability, ADHD, cortisol, inflammation, cytokines, stress, acute stress response, allostasis, anxiety

## Abstract

Attention-deficit/hyperactivity disorder (ADHD) is a prevalent and serious neurodevelopmental disorder characterized by symptoms of inattention and/or hyperactivity/impulsivity. Chronic and childhood stress is involved in ADHD development, and ADHD is highly comorbid with anxiety. Similarly, inflammatory diseases and a pro-inflammatory state have been associated with ADHD. However, while several works have studied the relationship between peripheral inflammation and stress in affective disorders such as depression or bipolar disorder, fewer have explored this association in ADHD. In this narrative review we synthetize evidence showing an interplay between stress, anxiety, and immune dysregulation in ADHD, and we discuss the implications of a potential disrupted neuroendocrine stress response in ADHD. Moreover, we highlight confounding factors and limitations of existing studies on this topic and critically debate multidirectional hypotheses that either suggest inflammation, stress, or anxiety as a cause in ADHD pathophysiology or inflammation as a consequence of this disease. Untangling these relationships will have diagnostic, therapeutic and prognostic implications for ADHD patients.

## 1. Background

ADHD is the most common childhood neurodevelopmental disorder and a public health concern, as about three-quarters of childhood cases persist into adulthood [1]. Further, a significant proportion of adults with ADHD did not have the disorder in childhood [2,3]. ADHD is characterized by symptoms of inattention and/or hyperactivity/impulsivity [4,5]. It has been associated with markedly worse school performance, independent of socioeconomic factors, in children [6,7], and with substantial impairment in social, occupational, and academic functioning in adults. Psychiatric comorbidities of ADHD include anxiety and mood disorders, tics, oppositional defiant disorder, learning disabilities, conduct disorder, and sleep disorders [8,9]. In ADHD adults, the rate of these comorbidities tends to increase with age, as well as with ADHD severity; the more ADHD symptoms, the more comorbid diagnoses have been identified [10]. Importantly, pooled data from twin studies have estimated ADHD heritability at around 76% [11,12]. It has been argued that ADHD includes multiple presentations, i.e., the predominantly hyperactive/impulsive presentation, the predominantly inattentive presentation, and the combined presentation [13]. Besides inattentive, hyperactive, and impulsive symptoms, ADHD is also characterized by emotion-related symptoms such as poor management of anger, irritability, and anxiety, with consecutive impairments in social relations, underachievement, and increased substance use [14,15,16,17,18]. Emotional dysregulation, or affective instability, is characterized by swift, high-intensity mood fluctuations marked by temporal instability and delayed recovery from dysphoria [19]. In fact, some authors suggest that ADHD could be included in the spectrum of emotion dysregulation disorders [20]. A meta-analysis of thirteen studies found that in the 2535 ADHD adults studied, emotion dysregulation levels were higher than in healthy controls. Negative emotional response and emotional lability, two emotion dysregulation dimensions, correlated with the severity of ADHD symptoms [21]. As will be discussed, anxiety and emotion dysregulation have been associated with stress, which, in turn, is implicated in ADHD pathophysiology and development. ADHD, being a neurodevelopmental disorder, presents some peculiar characteristics, as compared to other emotion dysregulation disorders. In particular, ADHD symptoms expose patients to stress early in childhood [22,23], and in turn, early-life stress in the form of various childhood traumas influences the development of the disorder [24,25,26,27].

If stress is associated with ADHD, additional emerging evidence also indicates alterations in inflammatory profiles and neuroendocrine stress reactivity in ADHD patients, possibly leading to immune dysregulation [5,28,29]. While several works have studied the relationship between peripheral inflammation and stress in affective disorders, fewer have explored this association in ADHD.

Therefore, since there is substantial evidence that (i) inflammatory pathways are dysregulated in some psychiatric illnesses, (ii), anxiety and stress are associated with inflammation and immune dysregulation, and (iii) anxiety and stress play a pivotal role in ADHD as in other emotion dysregulation disorders, we will review the existing literature on the relationship between stress, anxiety and immune dysregulation in ADHD, and how anxiety, stress, and acute stress, in particular, may induce a dysregulated inflammatory response in ADHD patients. Understanding stress and immune dysregulation from a pathophysiological viewpoint in ADHD may offer therapeutic interventions for treating the disorder and may help validate biomarkers to stratify ADHD severity. 

## 2. Relationship between Stress, Anxiety, and Inflammation

The definition of stress is intriguing and can be approached from different points of view, for example, by focusing on the response resulting from a certain negative stimulus in an individual. This approach may describe stress as a disruption of the normal psychological and/or physical homeostasis following external events that lead to a state of tension and pressure in the individual, or “an emergency state of an organism in response to a challenge to its homeostasis” [30]. Such external negative events are defined as stressors or, more simply, stress. More dynamically, stress may be characterized as “a particular relationship between the person and the environment that is appraised by the person as taxing or exceeding his or her resources and endangering his or her wellbeing” [31]. For classification purposes, stress might include chronic stress, acute stress, or early life adversities. Acute stress responses are typically easier to evaluate experimentally and can be shaped by early life or chronic stress. For such reasons, this review will focus mainly, but not only, on acute stress.

Interestingly, although they are distinct entities, a bidirectional interplay exists between stress and anxiety, which may be defined as “a temporally diffused emotional state caused by a potentially harmful situation, with the probability of occurrence of harm being low or uncertain” or “the anticipation of a future threat” [30]. Stress might lead to anxiety, but primary anxiety can also be considered a certain type of stressor [32]. The link between these two entities has a potential neurobiological basis, as noradrenergic projections to the basolateral amygdala from the locus coeruleus have been found to mediate anxiety following acute stress [30] and, in animal models, chronic stress may in turn favor anxiety-related behaviors through enhanced locus coeruleus reactivity [33]. 

Additionally, anxiety and stress have been increasingly linked with inflammation [34]. In healthy subjects, it has been consistently found that inflammatory markers (such as blood and salivary cytokines) are increased by acute stress [34,35,36]. Analogously, chronic stress has been associated with higher peripheral pro-inflammatory markers (e.g., plasma IL-6) [37]. Higher levels of pro-inflammatory peripheral cytokines have been identified both in models of anxiety disorders (tumor necrosis factor-α [TNF-α] and interleukin [IL]-1 β, IL-6) [38,39], as well as mood and anxiety disorders patients (c-reactive protein [CRP], TNF-alpha, IL-1, IL-6,) [40,41]. Recently, a metanalysis on 1077 anxiety disorders patients [42] showed that, excluding patients with a chronic organic illness, moderately higher concentrations of pro-inflammatory markers (again mainly IL1-β, IL6, and TNF-alpha) were present in patients’ blood, compared to healthy controls. No differences in anti-inflammatory cytokines were identified, even after correcting for comorbid depression. This strongly suggests that anxiety patients have a dysregulated immune response. Inflammation is, in turn, bidirectionally linked with disruption of the hypothalamic–pituitary–adrenal (HPA) axis, a crucial system of stress response regulation. As a matter of fact, both local and peripheral inflammatory mediators may impact the HPA axis, for instance, through a disrupted blood-brain barrier or the circumventricular organs [43]. On the other hand, a malfunctioning HPA axis reduces its physiological immunosuppressive and anti-inflammatory actions, facilitating a pro-inflammatory state [43]. Thus, anxiety is linked both with immune and stress response dysregulation, as shown, for instance, in a recent study on 2789 patients with depression or anxiety that showed that anxiety severity was associated with alteration of both immune and HPA indexes [44].

Finally, through imaging studies, inflammation has been shown to have an impact on connectivity in multiple brain structures and on some regions involved in anxiety (e.g., amygdala, anterior cingulate cortex, insula) in particular [45]. Similarly, a review on 399 mother-son dyads showed that an independent effect of stress on structural properties of cerebral white matter during different neurodevelopmental phases, i.e., pre-natal, post-natal, and adolescence [46]. 

## 3. Measures of Inflammation in Psychiatric Patients

Inflammation is increasingly recognized as a pivotal factor in a number of somatic pathologies and the interest in inflammatory biomarkers is growing (for example, see [47,48,49,50] for reviews on inflammation in neurology or cardiology). There is strong evidence suggesting immune dysregulation and a pro-inflammatory state in multiple psychiatric disorders [51,52,53,54,55,56,57], but the directional causality remains unclear [58]. Despite the rapid expansion of the field of immunopsychiatry, inflammatory markers are not usually measured in psychiatric clinical settings. In psychiatric research settings, inflammatory markers can be measured peripherally (e.g., in the saliva or in the plasma [36,59,60] or centrally (in cerebrospinal fluid [CSF] or brain tissue), and at rest or following certain tasks. 

A single saliva or blood test at rest can evaluate baseline levels of peripheral markers of inflammation and/or the state of hormonal pathways that modulate immune and inflammatory systems [60,61], such as the sympathetic–adrenal–medullary (SAM) system and the hypothalamic–pituitary–adrenal (HPA) axis (for a review, see [62,63]). Peripheral markers of inflammation include, for example, anti-inflammatory (such as IL4, IL5, IL10, transforming growth factor-β [TGF-β], C-C motif chemokine ligand-14) and pro-inflammatory (IL1, IL2, IL6, IL8, IL12, TNF-α, interferon-gamma, monocyte chemoattractant protein-1 [MCP1]) chemokines and cytokines, their receptors (e.g., soluble TNF-receptor-1 [sTNF-R1]), acute-phase proteins, blood immune cells, complement system components and antibodies. It should be pointed out that most of these measures of peripheral levels of inflammation are subject to inter- and intra-individual variability, for instance due to environmental factors such as smoking, obesity, diet, circadian rhythm, autoimmune diseases and infections. These, in turn, can be directly or indirectly linked to psychiatric disorders [64]. In fact, the interaction between pro-inflammatory cytokines and the brain is known, as well as their effects on neuroendocrine activity, cerebral structure, functions, and neurotransmission, to contribute to cognitive, behavioral, and emotional changes [65].

The aforementioned markers can also be evaluated before, during, and/or after a stressor, i.e., a situation or an activity that generates stress and/or anxiety in experimental volunteers. Typical examples of psychosocial stressors include mental arithmetic calculations or public speaking and are tested in the Trier Social Stress Test protocol (TSST), a human experimental gold standard [36,66]. Usually, the TSST is composed of an anticipation part and a test session. In the former, the subject is informed of the procedure that will follow, while in the test period, the participant has to perform surprise mental arithmetic and deliver a free speech in front of an audience. However, several adaptations exist, allowing for the TSST to be employed in combination with cerebral imaging, in group settings, with a virtual audience, or in children. The TSST is known to activate the HPA axis, although differences in stress responses may exist between sex, age, or time of the day [67,68].

Importantly, most of the markers that we mentioned may also be measured in post-mortem brain tissues and in CSF, reflecting central inflammation. Definitive evidence is lacking concerning the link between central and peripheral inflammatory markers, their pathophysiological relevance, and the potential directional causality. Despite this, peripheral inflammatory markers are easily obtained and have been shown to correlate in some cases with central markers. For instance, in major depressive disorder, CSF CRP strongly correlates with plasma CRP [69]. In psychiatric diseases, peripheral inflammatory markers have been studied more widely than central markers. Because of that, this review will mostly focus on them to discuss the relationships between stress, anxiety, and inflammation in ADHD, as has been done recently for emotion dysregulation disorders such as borderline personality disorder and bipolar disorder [70].

## 4. Stress and Inflammation in Patients with ADHD

Childhood trauma is associated with ADHD onset, severity, and persistence into adulthood [24,25,26]. Between 20% and 50% of children exposed to childhood trauma have clinical symptoms of ADHD [27,71]. Similarly, recent stress (i.e., stressful events in the previous year) positively correlates with ADHD symptoms in adult patients [72,73,74]. Interestingly, maternal stress during pregnancy increases the risk of subsequent ADHD in children [75,76,77]. 

In turn, ADHD, being a neurodevelopmental disorder, exposes patients to stress early in childhood. In fact, ADHD symptoms potentially expose patients to conflict, neglect, or physical and emotional abuse in social, schooling, and family settings [22,23]. It should be noted that physical trauma may be considered a type of stressor and, not surprisingly, traumatic head injury [78] is a risk factor for ADHD. However, ADHD children also have a higher risk of physical trauma and head injury compared to their healthy peers [79,80].

Moreover, as mentioned, anxiety disorders are a common comorbidity in ADHD and increase the severity of the condition. Further research is needed on treatment implications in ADHD with anxiety comorbidity, as contrasting results have been found on the efficacy of methylphenidate in this population [81].

The diversity of the studies that can be found on inflammation and ADHD indicates how much researchers and clinicians are looking into the origins of the disorder, but, at the same time, it highlights how far we are from pinpointing specific causes to the development of ADHD [82]. Discussing these studies is beyond the scope of this review, focusing on the relationship between stress, anxiety, and inflammation, but Table 1 gives an overview of the main evidence suggesting a link between inflammation and ADHD in human and clinical studies. As detailed in Table 1, cross-sectional observational studies, systematic reviews, and metanalysis have confirmed that ADHD is associated with autoimmune and inflammatory disorders, such as diabetes, psoriasis, asthma, allergic rhinitis and conjunctivitis, atopic dermatitis, maternal autoimmune disease, maternal stress, microbiome-dependent inflammation, polymorphisms in inflammation-related genes, and altered levels of immunological markers. These associations could be interpreted both as the result of an etiological role of inflammation in the development of ADHD or as the effect of a third variable causing both inflammation and ADHD. However, results from epidemiological and preclinical studies seem to support the former hypothesis [5,28,29]. For instance, there is evidence for a positive causal effect of body mass index (BMI) on ADHD [83]. This evidence comes from Mendelian randomization (MR), a technique that employs natural random distribution of genetic traits associated with certain risk factors to assess their link with a certain outcome, avoiding confounding variables in the absence of genetic pleiotropy [84]. A study applying MR on 55,374 ADHD patients from the Psychiatric Genomics Consortium found that subjects with genetic traits associated with higher BMI had a significantly higher risk of ADHD [83]. Adiposity is, in turn, causally associated with increased peripheral inflammation (as measured by CRP levels) [85]; thus BMI effect could be mediated by inflammation. Finally, another variable possibly contributing to inflammation in ADHD may be early life stress. In fact, we mentioned that ADHD patients might be exposed to early life adversities, which, crucially, are also a robust predictor of an increased pro-inflammatory status [65]. 

More to the focus of this review, we found four [59,89,103,104,105,106,107] original articles that examined peripheral inflammation in relation to stress and anxiety symptoms in ADHD patients. The investigation of inflammatory markers included pro-inflammatory cytokines such as IL1β, IL2, IL6, CRP, IFN-γ, and TNFa, and anti-inflammatory cytokines such as IL13. The activity of the HPA-axis was measured by means of the cortisol awakening response and dexamethasone suppression test (examining the sensitivity of the HPA-axis negative feedback). A study on ADHD children reported that ADHD, but not anxiety, symptoms were associated with increased levels of some inflammatory cytokines, such as IL16 and IL13 [103]. On the other hand, in a cohort study of 2307 adults with and without affective disorders, there was no association between ADHD symptomatology and IL6, CRP, or TNF-α [104]. In the same study, associations between HPA axis dysregulation and ADHD symptoms (including a significant correlation between hyperactivity/impulsivity symptoms and lower cortisol suppression) lost significance after adjusting for anxiety and depression [104]. Peripheral stress markers may also be used to stratify or identify children at risk for ADHD. Interestingly, among 417 healthy children younger than 8 years, those with higher salivary pro-inflammatory cytokine levels showed significantly stronger associations between ADHD symptoms and their caregivers’ perceived life stress scale scores [59]. 

Regarding measures of inflammation in relation to cortisol, definitive results have yet to be reached. In 44 ADHD patients of the inattentive subtype, there was a negative correlation between the level of pro-inflammatory markers (i.e., IL6 and TNF-α) and cortisol awakening response [89]. As mentioned above and as detailed in Table 2, the HPA axis is dysregulated in ADHD. We found nine original articles that examined the alteration of the HPA-axis in relation to ADHD symptomatology [105,106,107,108,109,110,111,112,113]. Several markers of the activity of the HPA-axis were investigated, including the cortisol awakening response, basal cortisol (evening or morning), dexamethasone suppression test, and cortisol concentration following laboratory stress (i.e., TSST). Past and current environmental stressors (such as acute life events, present adverse parenting conditions, and family conflicts) might also have an impact on HPA axis dysregulation as they were associated with higher cortisol awakening responses in ADHD children [110], for example. Strengthening the hypothesis that HPA axis impairment may be related to ADHD symptoms, in 102 healthy children, hyperactivity and impulsivity were associated with higher basal and acute stress-related HPA axis activity in boys but not in girls [107]. Interestingly, children with the inattentive subtype can have a reduced HPA axis reactivity to the TSST [112,113]. On the contrary, cortisol response to acute experimental stress, induced by an arithmetical ability test, has been found to be higher in ADHD adults (16 with the inattentive subtype, 1 with the hyperactive-impulsive subtype, 7 with the combined subtype) than in controls [106]. However, a study on 96 ADHD adults confirmed this finding only in the 38 patients with the inattentive subtype, who had higher levels of cortisol after TSST compared with the 96 combined ADHD patients [109]. Similarly, a study on 28 ADHD patients and 28 healthy controls showed that, among all participants, impulsivity levels correlated with high post-stress cortisol concentration, which was more common in ADHD patients. Higher post-stress cortisol was also associated with symptoms of depression and anxiety, which, as mentioned, may be possible confounders [105].

Furthermore, in ADHD adults, cortisol circadian levels present a significant delay of phase compared to controls [108]. It ought to be noted, however, that a study on 33 ADHD children and 33 age-matched controls failed to show a group-level difference in the cortisol awakening response [111]. It could be hypothesized that alterations in circadian rhythms are linked to HPA axis impairment in ADHD since short sleep duration has been associated both with higher morning cortisol [114] and with ADHD comorbid with affective disorders [115]. These associations are particularly relevant to stress and inflammation because the two have been linked to sleep disturbances [116,117]. 

In summary, the studies reviewed here support a multidirectional relationship between stress, anxiety, and inflammation with ADHD, but how these variables connect to each other has yet to be clarified (Figure 1). The major concern is that many of the studies give conflicting results on the dysregulation of inflammatory and stress-related mechanisms in ADHD [103,105,107,108,110,111,112,118] and that not all studies adjust for comorbidities such as anxiety and depression, which have both been associated with inflammation [40,41,42] or recognize anxiety as a mediating factor. Different subtypes of ADHD, as well as age or sex groups, might have a different relationship with stress reactivity and subsequent inflammation pathways. For example, a systematic review identified divergent genomic and metabolic findings between adult and childhood ADHD, suggesting differential etiological pathways depending on ADHD patients’ age [96]. In any case, building on these non-conclusive data, a vicious cycle hypothesis could be drawn, i.e., the symptoms of the disorder may themselves expose patients to chronic stress that leads to sleep disturbances, which further impair the HPA axis and exacerbate inflammatory dysregulation, possibly worsening ADHD symptoms. It is clear that further research is warranted.

## 5. Limitations

Due to the multifaceted nature of the research topic and for the sake of clarity, we performed a narrative review and not a systematic one, and we chose to focus on ADHD only, without including, for instance, papers discussing autism spectrum disorders or depression, although they might also be influenced by stress and inflammation [119,120,121,122]. 

## 6. Conclusions

A fascinating, albeit intricate, network of connections between ADHD, inflammation, anxiety, and stress emerges from this review. Although some experimental evidence and several hypotheses exist on this topic, the implications and the directions of such connections are still undetermined, and future research is warranted to pinpoint their precise interplay.

Hypothetically, ADHD and its symptoms (such as emotion dysregulation [123]) may expose a patient to a higher risk for stress, which, in turn, lowers the individual’s threshold for experiencing subjective stress, thus leading to abnormal stress reactivity, which is associated with inflammatory dysregulation. On the other hand, stress reactivity might be the *primum movens* that predisposes subjects to ADHD through the mediation of inflammation. In fact, inflammation can negatively impact brain structures, leading to neurodevelopmental disorders such as ADHD, possibly as a consequence of a dysregulated stress response.

However, an unknown confounding variable might exist, which may expose individuals to a higher ADHD risk, but also to inflammation, stress, and/or emotion dysregulation. Thus, the pro-inflammatory status observed in ADHD might be a cause, an effect, or just an epiphenomenon of this disorder. For example, some alimentary components and exposure to environmental toxic substances may be risk factors for both ADHD and a pro-inflammatory state/stress vulnerability, independently. Alternatively, the effect of these factors on ADHD risk may be mediated by inflammation. 

Indeed, it has been suggested that gluten-free or casein-free diet, food additives exclusion diet, and oligoantigenic diet are effective in reducing ADHD children’s symptoms, although study designs including food challenges and/or a reintroduction phase are needed [124,125]. If proven, this effect may be mediated by a reduction of subclinical allergic reactions or inflammation in response to these foods in ADHD children, who have been shown to be more prone to allergies and atopy. Similarly, free fatty acid supplementation may slightly attenuate ADHD symptoms [125], possibly through an anti-inflammatory effect [126]. Alternatively, or additionally, the role of gut microbiota has been hypothesized: some alimentary components may lead to pro-inflammatory status or oxidative stress through alterations in intestinal permeability and microbiome composition or function [124]. Gut microbiota is impacted by stress, and it is modulated by the HPA axis [127]. For instance, fecal microbial transplant reversed a pathological stress response in a rodent model [128]. Thus, higher exposure and/or sensitivity to stress due to ADHD may also lead to alterations in microbiota composition through disruption of the HPA axis, thus promoting a pro-inflammatory status and, possibly, further exacerbating stress reactivity. 

As mentioned, exposure to environmental toxic substances may also increase the risk of ADHD and related symptoms through inflammatory mechanisms. Children are likely more susceptible to exposure risks than adults due to their increased respiratory rate, smaller body size, and ongoing development [1]. In fact, exposure to environmental substances (e.g., nicotine in cigarette smoke, plasticizers and phthalates, persistent organic pollutants such as organophosphate pesticides, and metal such as Pb) has been associated with ADHD and ADHD symptoms [1]. Many of these substances are also believed to have an impact on the immune system [129,130], which may mediate this association.

An important point to be considered is the fact that anxiety is a common ADHD symptom, possibly mediating the aforementioned associations. As discussed, anxiety has been linked with inflammatory dysregulation, and we can hypothesize that increased levels of anxiety imply higher stress sensitivity, but the opposite may also be true [32]. In fact, stress cortisol response correlates with anxiety scores in healthy subjects [131]. Thus, another hypothesis might be that among all the models discussed in the paragraph above, anxiety may modulate the relationship between inflammation and stress. Anxiety may enhance stress-dependent inflammation or exacerbate a potential role of inflammation in dysregulating neuroendocrine response to stress, as has already been shown in anxiety disorders [40,41,42], and all these factors could lead to a higher risk of developing ADHD, as well. Importantly, while stress might lead to anxiety, these are different entities, as discussed in Section 3, with anxiety being an “anticipation of a future threat” and stress more a state of tension and pressure in the individual [30].

The identification of these relationships has important implications, whatever their mechanistic explanations or the direction of each association. Future studies should take into account the aforementioned environmental and clinical possible confounding variables, as well as potential sex, age, and subtype differences in stress reactivity in ADHD patients. Peripheral markers of HPA axis dysregulation and/or inflammation are easily measured, especially in relation to experimental anxiety and stress. On a hypothetical note, once better defined, these markers may become part of screening tests for subjects at high-risk for ADHD, or aid in diagnosis, for instance, possibly differentiating between ADHD subtypes. Should biomarkers of response to therapies be identified, they could assist in tailoring treatment and, speculatively, direct therapeutical interventions aimed at reducing inflammation and/or environmental exposure, as discussed above. Furthermore, the aforementioned relationships highlight the importance of identifying and treating comorbidities of ADHD. Thus, studies focusing on the association between inflammation, anxiety, and acute and chronic stress in ADHD are highly pertinent and might have quickly translatable clinical implications for assessing treatment response in patients, as well as diagnosis, prognosis, or risk evaluation in vulnerable individuals. 

## Figures and Tables

**Figure 1 biomedicines-09-01313-f001:**
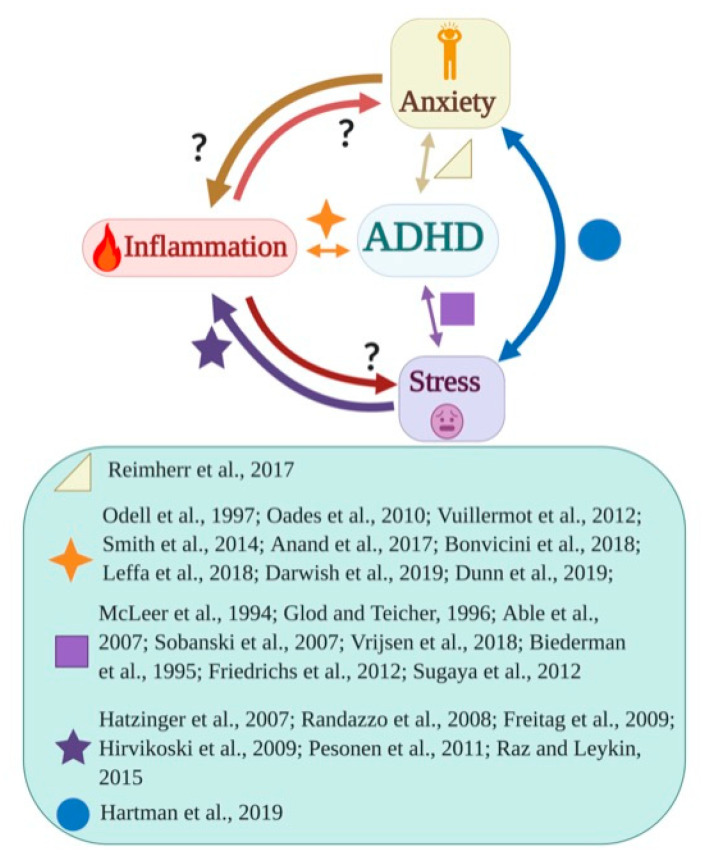
Links between inflammation, stress, and anxiety in attention-deficit/hyperactivity disorder (ADHD). The diagram depicts hypothetical relationships (indicated by arrows) between inflammation, stress, and anxiety in patients with attention-deficit/hyperactivity disorder (ADHD). Arrows connect couples of variables specifying the putative direction of the association between them in ADHD patients. Shapes close to a certain color-matched arrow are associated with main papers that support a link between the two specific variables connected by that arrow. Question marks indicate missing or insufficient evidence for a certain link, based on the reviewed articles. Figure 1 was prepared with BioRender.com (2021), accessed on 20 August 2021.

**Table 1 biomedicines-09-01313-t001:** Inflammation in attention-deficit/hyperactivity disorder (ADHD). Principal findings concerning inflammation in ADHD are recapitulated in the table. IL, interleukin; GWA, Genome-wide association.

Findings		Study Type	References
ADHD patients, including children, may have altered levels of immunological markers, such as: - increased cytokines (e.g., IL10 and IL6, although contrasting results exist on IL6) - auto-antibodies (e.g., anti-basal ganglia antibodies and anti-Yo antibodies targeting Purkinje cells)		Case-control observational study	Darwish et al.2019 [86]
		Comparative observational studies	Donfrancesco et al., 2016 [87] Toto et al., 2015 [88] Corominas-Roso et al., 2017 [89] Passarelli et al., 2013 [90]
ADHD is associated with autoimmune, atopic, and inflammatory disorders (e.g., diabetes, psoriasis, asthma, allergic rhinitis and conjunctivitis, atopic dermatitis/eczema)		Review	Chua et al., 2021 [91]
		Systematic reviews and Meta-analyses	Miyazaki et al., 2017 [92] Muskens et al., 2017 [93] van der Schans et al., 2017 [94]
Microbiome-dependent systemic inflammation might affect neurodevelopment and predispose to ADHD		Reviews	Bull-Larsen and Hasan Mohajeri, 2019 [95] Chua et al., 2021 [91]
Genetic polymorphisms in inflammation-related genes (e.g., antioxidant enzymes like superoxide dismutase, cytokine-related genes, and major histocompatibility complex) have been associated with ADHD		Systematic review	Bonvicini et al., 2018 [96]
		GWA analysis	Zayats et al., 2015 [97]
		Original papers	Odell et al., 1997 [98] Smith et al., 2014 [99]
Maternal pro-inflammatory factors	Maternal autoimmune, atopic, and inflammatory disorders (e.g., multiple sclerosis, type 1 diabetes, asthma, autoimmune thyroiditis) may increase the risk for ADHD in the offspring	Cohort epidemiological study	Nielsen et al., 2017 [100]
		Population-based nested case-control study	Instanes et al., 2017 [101]
		Systematic reviews	Fetene et al., 2017 [102] Han et al., 2021 [77]
	Stress during pregnancy increases the risk of ADHD in the offspring	Case-control study (intra-familial matched subject pairs)	Grizenko et al., 2012 [75]
		Prospective birth cohort study	Okano et al., 2018 [76]
		Systematic review	Han et al., 2021 [77]
	Maternal obesity, pre-eclampsia, smoking (before and during pregnancy), and low socioeconomic status increases the risk of ADHD in the offspring	Systematic review	Han et al., 2021 [77]

**Table 2 biomedicines-09-01313-t002:** Main studies linking either HPA-axis dysregulation or inflammation with attention-deficit/hyperactivity disorder (ADHD).

Findings	Study Design	Type of Marker	References
Elevated levels of IL16 and IL13 in ADHD children, independent of anxiety symptoms.	Cross-sectional study	Serum cytokines: IL1β, IL2, IL6, IL16, IL13, TNF-α, IFN-γ.	Oades et al., 2010 [103]
No association between ADHD symptomatology and IL6, CRP, or TNF-α. Loss of the association between HPA axis dysregulation and ADHD symptoms after adjusting for anxiety and depression.	Cohort study, cross-sectional	Plasma cytokines: IL6, CRP, TNF-α HPA-axis activity: Cortisol awakening response Dexamethasone suppression test Evening cortisol	Vogel et al., 2017 [104]
Children with higher salivary pro-inflammatory cytokine levels showed significantly stronger associations between ADHD symptoms and their caregivers’ perceived life stress scale scores.	Prospective Cohort study	Salivary cytokines: IL-6, IL-1β, IL-8, clustered into low, average, and high cytokine cluster groups by hierarchical cluster analysis	Parent et al., 2021 [59]
In 44 ADHD patients of the inattentive subtype, there was a negative correlation between the level of IL6 & TNF-α and cortisol awakening response	Cross-sectional study	Serum cytokines: IL6 & TNF-α Salivary cortisol awakening response	Corominas-Roso et al., 2017 [89]
In 102 healthy children, hyperactivity and impulsivity were associated with higher basal and acute stress-related HPA axis activity in boys but not in girls.	Prospective study	Salivary cortisol awakening response, salivary cortisol response to stress	Hatzinger et al., 2007 [107]
Among all participants, impulsivity levels correlated with high post-stress cortisol concentration, which was more common in ADHD patients. Higher post-stress cortisol was also associated with symptoms of depression and anxiety, which may be possible confounders	Cross-sectional study	Diurnal salivary cortisol and salivary cortisol in response to laboratory stress	Tatja Hirvikoski., 2009 [105]
Past and current environmental stressors were associated with higher cortisol awakening responses in ADHD children	Comparative study	Salivary cortisol awakening response	Freitag et al., 2009 [110]
Cortisol response to acute experimental stress was found to be higher in ADHD adults than in controls	Cross-sectional study	Salivary cortisol in response to laboratory stress	Raz, 2015 [106]

## Data Availability

Not applicable.

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
