# Peer review of "Inflammation, Anxiety, and Stress in Attention-Deficit/Hyperactivity Disorder"

_biomedicines, 2021, doi:10.3390/biomedicines9101313_

Round 1

Reviewer 1 Report

XCCCCCCCCCCCCCCCVXCXCCCVVVVVVVVVVVVVVVVVVVVVVVVVV

This article reviewed evidence of an interplay between stress, anxiety and immune dysregulation in ADHD. The theme is interesting enough, however, authors should address the following flaw.

My biggest concern is that I didn’t understand the ADHD-specific interplay between relevant factors very much. In other words, I suspect that ADHD located in the center of the Figure 1 can be replaced by any psychiatric conditions such as depression and ASD. I encourage authors to emphasize them more clearly.

Author Response

We thank the reviewer for this constructive viewpoint. While it is true that stress and inflammation play an important role in psychiatry in general (as mentioned in sections 3 and 4), ADHD has in fact some peculiar factors that make it particularly interesting and we have now tried to highlight them.

First of all, the interplay between stress anxiety and inflammation is less well studied in ADHD than in other disorders, such as MDD, for instance. Thus, we have now added at the end of the Background (Line 61, page 2):

“While several works studied the relationship between peripheral inflammation and stress in affective disorders, fewer have explored this association in ADHD.”

Secondly, one important ADHD-specific factor is its neurodevelopmental character, that makes maternal prenatal factors particularly relevant, as we discussed in Section 5 and at the end of Table 1.

We now additionally highlight other ADHD-specific factors in the Background (line 54, page 2) as follows:

“ADHD, being a neurodevelopmental disorder, presents some peculiar characteristics, as compared to other emotion dysregulation disorders. In particular, ADHD symptoms expose patients to stress early in childhood [22,23], and, in turn, early-life stress under the form of various childhood trauma influences the development of the disorder [24–27].”

And Section 5 reads (line 206):

“ADHD, being a neurodevelopmental disorder, exposes patients to stress early in childhood. In fact, ADHD symptoms potentially expose patients to conflict, neglect or physical and emotional abuse in social, schooling and also family settings [22,23]. It should be noted that physical trauma may be considered a type of stressor and, not surprisingly, traumatic head injury [73] is a risk factor for ADHD. However, ADHD children have also a higher risk of physical trauma and head injury compared to their healthy peers [74,75].”

Again, we recognize that ASD present some overlap with ADHD and also have close relationships with stress and inflammation. However, this review intersects numerous elements already (inflammation, stress, anxiety, and ADHD) and for the sake of clarity we deemed appropriate to focus on ADHD only. It would in fact be interesting to investigate similar relationships in ASD, as well, and therefore we added a new section discussing the limitations of our article, as follows (page 14, line 391):

     6. Limitations

Due to the multifaceted nature of the research topic and for the sake of clarity, we performed a narrative review and not a systematic one, and we chose to focus on ADHD only, without including, for instance, papers discussing autism spectrum disorders or depression, although they might also be influenced by stress and inflammation. ”

Reviewer 2 Report

First, I would like to note the importance and novelty of the topic, especially for the clinical approach of this pathology. The effort to make a synthesis of this vast and multidisciplinary subject is to be appreciated.

Regarding the objectives of this review I suggest to describe more precisely the markers that will describe stress and immune dysregulation associated with ADHD.

In the Materials and Methods section it would be useful to specify exactly the number of studies and patients included, including a short description of these samples.

In section 5 a table could be included with the description of the studies reported in the review and focused on the immune and inflammatory response in ADDH. Also in this section I would suggest a brief description of the markers used to measure these variables in patients with ADHD, maybe divided into categories.

For discussion section I would emphasize the importance of the topic in relation to the clinical approach and how these possible changes can help in the diagnosis and intervention process.

Author Response

We thank the reviewer for the very constructive comments, which we think greatly improved the article, as discussed below. In particular, we have detailed relevant studies and markers more precisely and exhaustively in Section 5 and in the new table, as suggested (see points below).

In the Materials and Methods section it would be useful to specify exactly the number of studies and patients included, including a short description of these samples.

We thank the Reviewer for this useful advice. While we did not aim at writing a systematic review due to the multifaceted nature of the topic, as specified in the Methods, we have now reported in the Methods the exact number of patients and studies that discussed the relationship between inflammation, anxiety or stress in ADHD, as follows (line 87, page 2):

“Four relevant studies reporting a relationship between ADHD symptomatology, anxiety/stress and inflammation were included with a total number of 2915 subjects. One study [98] focused on ADHD in children (35 ADHD children, 14 on medication and 21 controls). Another study focused on ADHD adults, including 44 ADHD patients of the inattentive subtype, 64 combined and 27 healthy controls, age ranging between 18 and 55 years [84]. A cohort study [99] included 2307 participants with and without affective disorders who completed ADHD symptoms questionnaire. Lastly, a study comprised 417 healthy children younger than 8 years who were evaluated for ADHD symptoms [54].”

In section 5 a table could be included with the description of the studies reported in the review and focused on the immune and inflammatory response in ADDH. Also in this section I would suggest a brief description of the markers used to measure these variables in patients with ADHD, maybe divided into categories.

We have inserted a second Table with the description of the studies as requested and a dedicated column for the details of the markers used in each of them (Table 2, page 11).

Furthermore, as suggested, besides the very general review of inflammatory and stress markers in Section 4, we have now reorganized this section and further described (line 253, page 9) the markers mentioned, as follows:

“The investigation of inflammatory markers included pro-inflammatory cytokines such as IL1β, IL2, IL6, CRP, IFN-γ and TNFa, and anti-inflammatory cytokines such as IL13. Activity of the HPA-axis was measured by means of the cortisol awakening response and dexamethasone suppression test (examining the sensitivity of the HPA-axis negative feedback).”

and (page 9, line 266)

“Peripheral stress markers may also be used to stratify or identify children at risk for ADHD. Interestingly, among 417 healthy children younger than 8 years, those with higher salivary pro-inflammatory cytokine levels showed significantly stronger associations between ADHD symptoms and their caregivers’ perceived life stress scale scores [54]. Regarding measures of inflammation in relation with cortisol, definitive results have yet to be reached. In 44 ADHD patients of the inattentive subtype, there was a negative correlation between the level of pro-inflammatory markers (i.e. IL6 and TNF-α) and cortisol awakening response [84]. We found nine original articles that examined the alteration of the HPA-axis in relation to ADHD symptomatology [100-104, 109-112]. Several markers of the activity of the HPA-axis were investigated, including the cortisol awakening response, basal cortisol (evening or morning), dexamethasone suppression test and cortisol concentration following laboratory stress (i.e., TSST).” (Section 5, page 9).

For discussion section I would emphasize the importance of the topic in relation to the clinical approach and how these possible changes can help in the diagnosis and intervention process.

We thank the Reviewer very much for raising this interesting point. The last section now reads (line 464, page 15):

“On a hypothetical note, once better defined, these markers may become part of screening tests for subjects at high-risk for ADHD, or aid in diagnosis, for instance possibly differentiating between ADHD subtypes. Should biomarkers of response to therapies be identified, they could assist in tailoring treatment and, speculatively, direct therapeutical interventions aimed at reducing inflammation and/or environmental exposure, as discussed above. Furthermore, the aforementioned relationships highlight the importance of identifying and treating comorbidities of ADHD.”